# Artificial Intelligence Reveals Distinct Prognostic Subgroups of Muscle-Invasive Bladder Cancer on Histology Images

**DOI:** 10.3390/cancers15204998

**Published:** 2023-10-16

**Authors:** Okyaz Eminaga, Sami-Ramzi Leyh-Bannurah, Shahrokh F. Shariat, Laura-Maria Krabbe, Hubert Lau, Lei Xing, Mahmoud Abbas

**Affiliations:** 1AI Vobis, Palo Alto, CA 95054, USA; 2Department of Urology, Pediatric Urology and Uro-Oncology, Prostate Center Northwest, St. Antonius-Hospital, 33705 Gronau, Germany; 3Department of Urology, Comprehensive Cancer Center, Medical University of Vienna, 1090 Vienna, Austria; sfshariat@gmail.com; 4Department of Urology, University Hospital of Muenster, 48419 Muenster, Germany; 5Department of Pathology, School of Medicine, Stanford University, Stanford, CA 94305, USA; hlau@stanford.edu; 6Department of Pathology, Veterans Affairs Palo Alto Health Care System, Palo Alto, CA 94304, USA; 7Department of Radiation Oncology, School of Medicine, Stanford University, Stanford, CA 94305, USA; 8Department of Pathology, University Hospital of Muenster, 48419 Muenster, Germany

**Keywords:** deep learning, digital biomarker, bladder cancer, FGFR3, WSI, risk stratification, histology images

## Abstract

**Simple Summary:**

This study developed an interpretable scoring system using artificial intelligence and bladder tissue images. It identified two distinct risk groups with different outcomes in high-grade bladder cancer. The scoring system was associated with various molecular features and gene mutations. This system can save shared clinical decision making and cost by identifying patients who need further molecular testing.

**Abstract:**

Muscle-invasive bladder cancer (MIBC) is a highly heterogeneous and costly disease with significant morbidity and mortality. Understanding tumor histopathology leads to tailored therapies and improved outcomes. In this study, we employed a weakly supervised learning and neural architecture search to develop a data-driven scoring system. This system aimed to capture prognostic histopathological patterns observed in H&E-stained whole-slide images. We constructed and externally validated our scoring system using multi-institutional datasets with 653 whole-slide images. Additionally, we explored the association between our scoring system, seven histopathological features, and 126 molecular signatures. Through our analysis, we identified two distinct risk groups with varying prognoses, reflecting inherent differences in histopathological and molecular subtypes. The adjusted hazard ratio for overall mortality was 1.46 (95% CI 1.05–2.02; z: 2.23; *p* = 0.03), thus identifying two prognostic subgroups in high-grade MIBC. Furthermore, we observed an association between our novel digital biomarker and the squamous phenotype, subtypes of miRNA, mRNA, long non-coding RNA, DNA hypomethylation, and several gene mutations, including FGFR3 in MIBC. Our findings underscore the risk of confounding bias when reducing the complex biological and clinical behavior of tumors to a single mutation. Histopathological changes can only be fully captured through comprehensive multi-omics profiles. The introduction of our scoring system has the potential to enhance daily clinical decision making for MIBC. It facilitates shared decision making by offering comprehensive and precise risk stratification, treatment planning, and cost-effective preselection for expensive molecular characterization.

## 1. Introduction

Bladder Cancer (BC) is the tenth most common cancer in the United States, mostly affecting people older than 55 years. Bladder cancer (BC) exhibits a gender disparity, affecting men approximately four times more frequently than women [1]. Furthermore, BC encompasses a broad spectrum of disease behavior, ranging from a slow-growing non-muscle-invasive form (NMIBC) to a highly aggressive muscle-invasive variant (MIBC). Although most BC patients are diagnosed with NMIBC, up to 25% of BC are identified as MIBC with substantial risk for mortality [2]. BC cases with stage I or II show a 5-year relative survival rate of 96% or 70%, respectively, whereas 38 of 100 cases with stage III will survive 5 years; cases with stage IV have the poorest survival outcome with 6% of a 5-year relative survival rate. Moreover, BC reveals distinct multilevel molecular subtype profiles associated with prognosis and treatment responses [3]. However, determining multilevel molecular subtype profiles (i.e., protein expression, gene mutation, mRNA, DNA methylation, and miRNA) requires a complex and expensive infrastructure likely unavailable in most cancer centers worldwide. Therefore, a cost-effective solution could ideally help to manage the patient selection according to their risk of having progressive cancers or to identify cases likely to benefit from certain treatment regimens.

Recent studies revealed the potential of deep learning (DL) to predict a new generation of digital biomarkers for detection, prognosis, molecular signature, and treatment response in different cancers, including bladder cancer [4,5,6]. For instance, Woerl et al. reported the potential of DL to forecast the molecular subtypes of MIBC by analyzing hematoxylin and eosin (H&E) slides [7]. As a proof-of-concept, Mundhada et al. have shown the DL capability to distinguish low-grade from high-grade histology [8]. Zheng et al. purposed a DL framework to predict survival from histology images with BC [9]. While deep learning holds immense potential, addressing certain tendencies that have arisen within its application is essential. Specifically, a majority of prior research treated confidence scores as equivalent to probability scores, disregarding the well-recognized problem of overconfidence in deep learning models [10,11]. Furthermore, these studies have not provided a feasible means of interpreting whether the feature distributions in the latent feature spaces reflect alterations in histological patterns that contribute to the prediction scores.

Given the limitations of previous studies, our hypothesis posits that morphometrical patterns observed at the histological level are indicative of prognostic confidence scores, which are then associated with omics signatures specific to advanced bladder cancers. Our primary objective is to identify prognostic subgroups that reveal associations with molecular subtypes, utilizing histology images, including bladder cancers and weakly supervised learning. The major contribution of the current work is to provide a novel strategy that facilitates the development of interpretable prognostic scores derived from a collection of mixed histology patterns associated with molecular subtypes and potential treatment options for bladder cancers.

## 2. Methods

### 2.1. Survival Modeling

#### 2.1.1. Data

Complete data were available for 113 patients diagnosed with urothelial carcinoma of the bladder (BC) from the Prostate, Lung, Colon, and Ovarian Cancer Screening (PLCO) trial. PLCO is a randomized controlled trial aimed to determine whether certain screening exams reduce mortality from prostate, lung, colorectal, and ovarian cancer (NCT00339495) [12,13]. Although this trial did not screen for BC, it tracked diagnoses of BC during the trial period. Briefly, 154,900 participants from the general population aged 55 through 74 years were enrolled between 1993 and 2001 [14]. Only subjects without a history of prostate, lung, colorectal, or ovarian cancer were enrolled. Cancer diagnoses were confirmed by retrieving results and information from medical records and the cancer registry system. This study used a linkage with the National Death Index to extend mortality follow-up to a maximum of 19 years after randomization [15]. During the study follow-up period, 1430 cases of BC were diagnosed, from which the PLCO study organizer randomly selected 285 cases to scan representative whole slides with samples containing BC. All samples were originally obtained through transurethral resection of bladder tumors.

After excluding the slide images of cases with missing follow-up information, a total of 196 H&E-stained slides of the bladder cancer cohort were available from nine U.S. centers and digitally scanned at 40× objective magnification (one pixel corresponds to ~0.2532 µm) using a Leica Biosystems device (Wetzlar, Germany) and stored in SVS format.

We split these images, as the development set, into a training set, optimization set, and validation set by institutions to prevent overlapping between these sets; cases of a center having the largest portion in our cohort were selected for the training set, the center with the smallest portion was considered for optimization set and the remaining centers for the validation set. Figure 1 summarizes our framework for developing the digital biomarker for mortality.

#### 2.1.2. Image Preprocessing

The rectangle boundary of the tissue area was estimated after thresholding the gray color version of the thumbnail image (1× magnification) for each image and upscaled to correspond to 40× magnification. After that, the tissue area was divided into 2048 × 2048 pixels (px) tiles, and tiles mostly (>50% of the tile pixels) matching the white background colors were excluded. The resulting tiles were downsized to 512 × 512 px (~10× objective magnification). Each tile originated from the same patient and was labeled for the binary cancer-specific death status (CSD) on the death certificate.

#### 2.1.3. Model Development

The current study applied the neural architecture search (NAS) algorithm for PlexusNet [16] and the training set to determine the optimal model architecture for CSD prediction. Here, we used the grid search and an abstract search space covering the type of block (i.e., attention block, ResNet, or inception block), depth (i.e., how often to repeat the blocks), and the branching factor (i.e., number of multi branches in the network) of the convolutional neural network and the transformer inclusion, resulting in the examination of 1296 models with different architecture configurations (Table 1). In addition, we applied the widely accepted optimization algorithm “ADAM” with the standard hyperparameter configuration and the cross-entropy loss function to train each model for one epoch. For the NAS, the batch size was set to 64 patches and the learning rate to 1 × 10^−3^. To optimize the computational efficiency of NAS, we employed a downsizing technique from our previous work, reducing the patches to a 32 × 32 pixel dimension [16]. This approach allows us to focus computational resources on smaller patches, reducing complexity while extracting meaningful information. The downsized patches balance between computational efficacy and the ability to explore diverse architectural designs, streamlining the NAS process for large-scale experiments and real-world applications [16]; the two-fold cross-validation was applied to train and evaluate each model for balanced classification accuracy. Finally, the final model architecture with the highest average performance on two-fold cross-validation was selected.

The resulting model was then trained on the whole training set with 512 × 512 px patches until convergence. During model training, we set an early stopping algorithm (stop training when the loss values on the optimization setting are not improved for ten epochs) to mitigate the model overfitting; Adam with weight decay was applied as instructed by the authors for model training while the learning rate was set to 1 × 10^−4^. The binary patch label was randomly smoothed with +/− 0.25 to moderate the model overconfidence in addition to model overfitting and to improve the model calibration. The image augmentation was applied and included random rotation, flipping, clipping, and color space augmentations, as described previously in the image preprocessing section. For each epoch, we validated the model performance on the optimization set at the patient level. Here, we measured the average confidence scores for CSD on all patches for each patient and the discriminative accuracy for CSD prediction using a time-dependent area under the receiver operating characteristic curves (AUROC) and c-index at the case level.

We applied the validation set to validate the case-level model accuracy at the patient level and to visualize the feature space of the last convolutional layer (not the global pooling) using t-SNE (t-distributed stochastic neighbor embedding). We then clustered the feature spaces according to the deciles of CSD prediction to visualize the correspondence between CSD prediction and feature space. After that, we determined two deciles based on the feature clusters. The first decile cluster (reference decile, *r*) shows a feature space dominant for negative patches, whereas the second decile cluster corresponds to the median decile (*m*).

After finding deciles *r* and *m*, we developed an algorithm to estimate the CSD score for each case as follows:(1)We first calculated the patch frequency for 10 bins with equal width (histogram) at the case level. The bin width was calculated for each case using Equation (1).
(1)$$bin\;width=\left(s_{max}-s_{min}\right)/\sqrt{10}$$
where *s_max_* is the maximum CDS score, and *s_min_* is the minimum CSD score for each case.

(2)Secondly, we applied the maximum normalization to the patch frequencies, including *D_r_* and *D_m_*, to achieve a value range between 0 and 1 for all bins. Third, we estimated the unadjusted CSD score (*S_u_*) using the following equation:


(2)
$$S_{u}=D_{m}-D_{r}$$


(3)Since out-of-distribution data may have a different frequency distribution than the development set, we introduced the following algorithm to adjust the CSD score estimation without having the ground truth:
(a)Calculate the mean µ of *S_u_*;(b)Calculate the median μ_1/2_ of (*S_u_* − µ);(c)Adjust the scores by µ and μ_1/2_ according to the equation:



(3)
$$S_{CSD}=-\left(\mu \;+{\;\mu }_{1/2}\right)+D_{m}-D_{r}$$


We also applied thresholding to *S_CSD_* to define a binarized risk category. The threshold (*T*) was determined using the following equation:T = µ_0_ + 1.05 σ_0_ + **α**(4)
where σ_0_ is the standard deviation of *S_u_*, σ_0_ and µ_0_ were calculated on development set, and **α** is the correction factor that counts the difference between µ_0_ and the mean of out-of-distribution cohort (µ_c_) and can be expressed as
**α** = µ_0_ − µ_c_(5)

Since the bin range differs from case to case by the CSD score range, we asserted that the median for the case-wise midrange of CSD scores for $D_{r}$ (MR = 0.17; interquartile range, IQR: 0.16–0.18) and $D_{m}$ (MR = 0.41; IQR: 0.37–0.42) was comparable between the development and out-of-the distribution cohort (external validation) to ensure the generalization of binning with equal width.

### 2.2. Evaluation

#### 2.2.1. Data

We obtained 457 H&E-stained whole slide images from The Cancer Genome Atlas (TCGA)—Urothelial Bladder Carcinoma cohort [17], from which 412 images included survival information. This TCGA cohort contains genetic, demographic, and clinical outcome data for various cancers, and this data is made publicly available through their online platform (NCI Genomic Data Commons). The TCGA study for bladder cancer has received contributions from 36 institutions worldwide. The sources of bladder cancer tissue specimens were radical cystectomy (RC) specimens. The slides with bladder cancer tissue specimens were digitally scanned at 40× objective magnification (one pixel corresponds to ~0.2532 µm on average) using a Leica Biosystems device (Wetzlar, Germany) and stored in SVS format. Clinicopathological and follow-up information was available at the case level. We also applied the same image preprocessing strategy and scoring system described earlier to this cohort. All images with available molecular profiles and clinicopathological and follow-up information were considered. Each case corresponded to a single whole-slide image.

#### 2.2.2. Prognosis

We assessed the prognostic value of our novel risk group using the univariate and multivariate Cox proportional hazards models. In multivariate analysis, cancer-stage grouping and age at diagnosis were added to adjust the hazard ratio for the novel risk group. The outcome was the overall survival (OS) from the diagnosis, as the TCGA dataset is highly qualitative and widely used for overall survival analyses in cancer research [18]. Patients lost to follow-up were censored at the date of the last contact.

#### 2.2.3. Association with Familiar Molecular Signatures of Bladder Cancer

We evaluated seven histopathologic (e.g., squamous phenotype) and 126 molecular signatures (e.g., the mutation in FGFR3 and molecular subtypes) investigated by the TCGA study [17] in bladder cancers (see the signature list in the Appendix A) for their association with the categorized risk score groups. In addition, for any significant signatures with more than two categories, we performed post hoc comparison analyses to determine which categories significantly differ between the novel two risk groups.

### 2.3. Metrics, Statistics and Software

We applied the time-dependent AUROC at the fifth follow-up year [19] and univariate and multivariate Cox regression analyses to assess our novel scoring system on the development set before the external validation.

The classification and accuracy of prognosis were quantified with AUROC and Harrel’s c-index [20,21]. The goodness of fit was measured according to the Akaike information criterion (AIC) and Bayesian information criterion (BIC), where the lower the value, the better the model fit [22,23,24]. Finally, Kaplan–Meier survival estimates were applied to approximate the survival probability for our novel risk classification.

The chi-square tests were performed to determine whether there is an association between categorical variables (n × m contingency tables). In contrast, the Fisher test was applied to estimate the odd ratios and assess 2 × 2 contingency tables. Finally, we used the Wilcoxon Rank Sum Test to assess the differences in a numerical variable between the novel risk groups.

The comparison analyses for categorical signatures include repeating the Fisher test for each signature category as one-versus-other and the significance determination for each comparison test according to the Benjamini-Hochberg (B-H) procedure [25]. Here, the critical value was calculated for each comparison test after the *p*-values of comparison tests were ranked from low to high. The following equation was used to estimate the critical value at a false discovery rate (FDR) of 0.20:Critical value = rank/(number of comparisons) × 0.20(6)

A comparison test is deemed significant according to the last *p*-value lower than its critical value. The Pearson Correlation coefficient estimated the correlation between two numerical variables, while the Kendall rank correlation coefficient (τ) was estimated to measure the ordinal association between one numerical variable and one categorical variable or between two categorical variables [26,27]. VIF (variance inflation factor) was used to assess the multicollinearity in the COX regression model [28].

Model development and analyses were performed using Keras 2.6 [29], TensorFlow 2.10 [30], Python™ 3.8, and the R statistical package system (R Foundation for Statistical Computing, Vienna, Austria). All statistical tests were two-sided, and statistical significance was set at *p* ≤ 0.05 for prognosis or *p* ≤ 0.10 to consider molecular or histopathologic signatures for comparative analyses.

## 3. Results

### 3.1. Survival Modeling

Table 2 summarizes the cohort description of the development set. We found no significant difference in the cohort characteristics between the subsets (i.e., training, optimization, and validation sets). We considered the diverse BLC pathologies (not limited to muscle-invasive bladder cancer) to increase the likelihood of capturing differential histopathological patterns by our model for prognosis. In alignment with the literature, 77% of training set cases were non-muscle invasive BC and representative of the population. The optimization set was utilized to fine-tune the model, enabling it to distinguish between non-lethal and lethal patches while considering the various WHO Grades that exhibit heterogeneous patterns. Using a small sample size for optimization allowed the domain expert to manually review the predicted patch classes and streamline performance optimization accordingly. The validation set had a balanced distribution of NMIBC and MIBC cases, and G1/2 and G3 cases, thereby minimizing the effect of sampling bias.

The Neural Architecture Search (NAS) examined 1296 PlexusNET architecture configurations (duration: ~12 h) [16] and suggested a shallow model (model configuration: VGG D6L2J1F2 + transformer and global average pooling; these parameters regulate the design of the model architecture and the model scaling) having only 23,783 parameters and 20 fully connected representation features as the best model configuration for cancer-specific death (CSD) prediction. The Levene test indicated a significant difference in 18 out of the 20 two-dimensional feature maps between the patches derived from patients who died due to bladder cancer and those who survived. In other words, the feature maps were found to be unequal or dissimilar between the two groups, indicating the extraction of significant feature representation for CSD from histology images (Figure 2).

Following the instructions provided in the Section 2 to derive a risk score from histology images, we visualized the feature space and determined the feature subspaces using the prediction deciles. The t-SNE visualization of the feature space showed that the prediction deciles sorted feature points, and the evaluation of the corresponding patch images confirmed the differences in histopathology appearance according to the deciles (Figure 3). Therefore, based on the t-SNE feature visualization and the assessment of the histopathology appearance, the second decile (D2) and the fifth decile (D5) met the selection criteria described in the Section 2.

At the patient level, the risk score was prognostic for cancer-specific mortality (HR: 8.0; 95% CI: 1.4–46.1; z: 2.332; *p* = 0.0197). The 5-year AUC was 0.772 ± 0.04. The multivariate Cox regression analysis further strengthened the independent prognostic significance of our novel risk score, even after adjusting for age at diagnosis and tumor grade. Including our novel scoring system in the analysis offered an alternative approach for assessing histopathological characteristics that is distinct from tumor grade (Table 3).

The Kaplan–Meier Curve revealed that the risk score (categorized) delivered two distinctive risk groups (*p* = 0.014), as shown in Figure 4. The median survival for the high-risk group was achieved between 204 (17 years) and 216 months (18 years) after the initial diagnosis.

### 3.2. Prognosis for Muscle-Invasive Bladder Cancer

Table 4 summarizes the cohort description of the external validation set. The vast majority of cases included high-grade MIBC. The distribution of the risk scores around the cohort-specific threshold (T = 0) is shown in Figure 5. The categorization of the risk score was driven by the dominance of either D2 or D5 in each case, and D2 and D5 were associated with distinct histopathologic patterns of bladder cancers in the TCGA cohort (Figure 6).

We found that the risk groups are prognostic for overall survival on the external validation set (HR: 1.46; 95% CI: 1.05–2.02; z: 2.23; *p* = 0.03). The multivariate Cox regression analysis showed that risk groups are, in addition to the pathologic stage and age at diagnosis, independent prognosticators for overall survival as well (Table 5). The multicollinearity for these covariates was negligibly small (VIFs < 2).

The Kaplan–Meier curve and the log-rank test indicate that the risk groups were statistically distinct (*p* = 0.037), as shown in Figure 7. Both risk groups reached the median overall survival, but at different time points (~30 months for high-risk vs. ~60 months for low-risk); the high-risk group reached the median survival ~2.5 years earlier than the low-risk group for muscle-invasive bladder cancers. Figure 8 provides the Kaplan–Meier curve for the stages of bladder cancer for comparison.

### 3.3. Association with Molecular Signatures of Bladder Cancer

We identified molecular and pathologic signatures significantly associated with the risk groups at case level, as shown in Table 6. Specifically, the TCGA clusters for miRNA, mRNA, lncRNA, and DNA methylation were associated with our novel risk groups. In addition, multiple mutations, including TSC1, FGFR3, and ERBB3, occurred differently between the novel risk groups.

The luminal papillary cluster was associated with the low-risk group, whereas the basal/squamous cluster and the neuronal cluster were associated with the high-risk group (Table 7). Moreover, cluster 2 for DNA hypomethylation is associated with the high-risk group; in contrast, cluster 4, with lesser DNA hypomethylation than cluster 2, was associated with the low-risk group. At the long non-coding RNA level, cluster 3 was frequently seen in the low-risk group and cluster 4 in the high-risk group. At the miRNA level, cluster 3 was more frequent in the low-risk group, and cluster 4 was common in the high-risk group.

The low-risk group included 72% of the TSC1 mutation (28 of 39 TSC1 mutations) or 67% of the ERBB3 mutation (30 of 45 ERBB3 mutations) in bladder cancer (Table 8 and Table 9). The odd ratio of TSC1 mutation was 0.36 (95% CI: 0.15–0.76; *p* = 0.004), and the odd ratio of ERBB3 was 0.46 (95% CI: 0.22–0.91; *p* = 0.0179) for high-risk groups.

The true positive rate of our low-risk group was 65% for FGFR3 mutations (Table 10) with an AUC of 0.593 (95% CI: 0.55–0.69). The odd ratio for FGFR3 mutation in the high-risk group was 0.49 (95% CI, 0.27–0.87; *p* = 0.0102). The high-risk group included 63.3% of the squamous pathology. The supplementary section provides different results for significant signatures.

## 4. Discussion

In this study, we developed and externally validated an AI-based algorithm that stratifies muscle-invasive bladder cancer by mortality risk directly from histology images. Moreover, our novel risk groups can reveal which histopathological pattern is dominant in tissues with bladder cancers. Our approach is feasible thanks to the intuitively well-sorted feature space generated by weakly supervised learning. This property has made it possible to discretize the feature space into ten small segments organized decile-wise, allowing us to evaluate the histopathological patterns for each prediction decile.

Earlier studies in bladder cancer applied deep learning to infer staging [31], grade [32,33], recurrence risk [34], FGFR3 mutation [35], and specific molecular subtypes [7] from histology images. Although some previous studies examined the prediction of molecular targets, the current study found that prognostic histopathological patterns for bladder cancer are rather associated with multi-omics profiles (i.e., transcriptomic, genomics, and epigenomics); these multi-omics profiles are already covering the specific molecular subtypes and the FGFR3 mutations investigated earlier, and we have shown that the accuracy of our risk groups for FGFR3 mutation is similar to the previous report, signifying the impact of multi-omics profiles as confounding factors on the results of earlier studies. In support of our findings, the BLCA-TCGA study (molecular characterization of bladder cancer) revealed that the molecular subtypes and signatures are linked with each other and distinct histopathologic patterns (e.g., papillary, basal/squamous) were connected with omics profiles that are prognostic and have different therapeutic targets [3,17]. A comparable study in Lung cancer reported that omics features are predictive of histology patterns as well [36].

Although multiple studies identified the detection potential of single mutations or specific molecular subtypes from histology images [37,38,39,40,41], the histopathological appearance is mainly driven by a collection of multifaceted molecular modulations and reflects the cancer malignancy and survival. Subsequently, establishing a direct association between a single molecular signature and histology images must be inadequate, given other confounders for bladder cancers.

Our novel risk groups are linked with therapeutic targets like FGFR3 (erdafitinib) [42], ERBB3 (afatinib) [43], PI(3)K (LY294002, other mTOR inhibitors) [44,45], and TSC1 (nab-sirolimus, study no.: NCT05103358) [3] as well as with female gender-biased gene mutations like KDM6A mutation (a histone lysine demethylase) [46]. Accordingly, our novel risk group holds a potential clinical utility in pre-screening for mono- and combinational target therapies (Figure 9). This potential will be more evident once prospective randomized studies to validate the clinical utility of our approach for patient selection in the real-world clinical setting are available.

A detailed examination of the multi-omics profiles associated with our risk groups reveals unique molecular regulatory profiles at the microRNA, lncRNA, and DNA methylation levels. We found that the low-risk group is linked with molecular subtypes with good survival for coding and non-coding RNAs or DNA methylation. These multi-omics subtypes are associated with papillary tumors, high FGFR3 mutations and miR-200 levels, and low Epithelial–Mesenchymal Transition (EMT) scores, CD274 (PD-L1) and PDCD1 (PD-1) level [17]. In contrast, the high-risk group is linked with molecular subtypes with poor survival for coding and non-coding RNA, which are further associated with lymphocyte infiltration, the high expression of CIS (carcinoma in situ) signature genes, CD274 (PD-L1) and PDCD1 (PD-1) levels, high TP53 mutations and EMT scores [17]. The high-risk group is additionally linked to cluster 2 for DNA hypomethylation, which has more DNA hypermethylation signals (more gene inactivation) than cluster 4, which is linked with the low-risk group [17]. Our data further facilitates deriving a hypothesis that the low-risk group, with favorable multi-omics profiles, is likely more responsive to different targeted therapies than the high-risk group, and the high-risk group may benefit from immune checkpoint inhibitors (i.e., anti-PD-1 or PD-L1); our data also suggest that epigenetic therapy could be a potential therapeutic option for our high-risk group. Figure 9 summarizes each risk group’s molecular characteristics and potential treatment options.

Comparable studies utilized activation maps or tiles with top scores to interpret the model inference. However, the trustworthiness of activation maps could be more questionable as deep neural network classifiers have an opportunistic nature, and the existing saliency methods inherit a high risk for misinterpretation, limited reproducibility, and sparse visualization [47,48]. Moreover, it should be considered that tiles with top scores ignore the variance in histology patterns between two categories after thresholding predictions, as evident by our data on the correlation between histology patterns and prediction deciles.

We applied the neural architecture search to achieve a data-driven architecture design with a better trade-off between accuracy, interpretability, and model complexity. In our study, only 20 feature representations (i.e., the 2D feature maps of the last convolutional layer) are sufficient to derive accurate predictions from histology images and correspond, for example, to 4% of feature representations of ResNet18 [49] (i.e., 512 features), an off-the-shelf model commonly used in medical imaging research. Reducing the feature representation is associated with a better computation cost for downstream analysis and improved human interpretation of these features. Moreover, our approach helps visualize and analyze three-dimensional representative features that preserve topological information at reasonable computation costs (e.g., analysis of 8,000,000 data points required ~30 min using parallel computing). In contrast, comparable studies that utilized off-the-shelf models are limited by extremely reduced feature granularity (1D) with loss of topological information for downstream analysis, given the high computation cost to analyze a large number of 3D representative features that these models have. Accordingly, comparable studies excluded the most information from the feature representation to conduct downstream analysis. In contrast, our approach preserves the high granularity of the feature representation for downstream analysis and consequently improves the interpretability of our AI model.

Despite the strengths of our study, it is essential to acknowledge certain limitations. Firstly, using slide images introduces potential variability in image quality due to factors such as diverse scanning technologies, staining variations, and image artifacts. These variations can introduce inconsistencies that may impact the accuracy and reliability of image analysis and interpretation. Nevertheless, we took measures to mitigate this concern by using PlexusNET to address the domain shift [16], conducting a comprehensive manual review involving domain experts and validating our findings on multicentric datasets. Additionally, we employed feature visualization techniques to identify the potential impact of artifacts and reviewed for the staining variations on the selected histological patterns. Secondly, it is crucial to recognize that TCGA slide images offer a glimpse of a specific tumor region or patient sample, which may not fully capture the complex intra- and inter-tumor heterogeneity. Tumors can exhibit spatial and molecular heterogeneity, resulting in significant variations between different regions within the same tumor or among tumors of the same type. Analyzing only a subset of slide images may provide an incomplete representation of tumor characteristics. Nonetheless, it is noteworthy that the TCGA and PLCO study followed good research practices, aiming to select the most representative samples from each patient according to the existing technical feasibility. Moreover, the quality of survival data of TCGA was validated for overall survival analyses [18]. The good research practices and the data quality help mitigate this limitation to some extent. It is important to emphasize that TCGA slide images, obtained through the TCGA project, do not directly correspond to the specific sampling areas used for molecular examination. These images are prepared using Hematoxylin and Eosin (HE) staining, a common technique for histological analysis. In contrast, molecular examinations and profiling involve separate samples or portions of the tumor that undergo different processing steps. TCGA employs distinct protocols for various analyses, including genomic, transcriptomic, and proteomic profiling, which are not directly applied to the same tissue sections used for generating slide images. These protocols often utilize specialized techniques, such as DNA sequencing or protein expression analysis, requiring separate tissue preparation and processing. Hence, it is crucial to note that TCGA slide images, while they provide valuable histological information, do not directly correlate with the specific regions of the tumor that underwent molecular examination. Rather, they serve as representative snapshots of the tumor’s morphology and architecture, offering valuable context for researchers analyzing the genomic and molecular data obtained from the TCGA project. We preferred slide images with formalin-fixed paraffin-embedded (FFPE) tissues as this approach offers standardized staining and more reliable histology images. In contrast, the process of preparing and staining frozen tissue slides are demanding and often result in associated artifacts; freezing can cause structural changes and cellular damage, while its staining consistency can be challenging due to variations in tissue quality and protocols [50,51,52]. Finally, histology images from frozen sections are also snapshots, contrary to a common misconception that assumes these images are direct complements to the entire TCGA samples.

The current study introduces a novel AI-based risk grouping system for survival derived from bladder cancer H&E slides. We show the linkage between our risk groups and multi-omics profiles for muscle-invasive bladder cancers. We highlight the concerns with predicting single molecular signatures (e.g., FGFR3) from histology images. While our approach has been rigorously tested and validated in the context of bladder cancer, its applicability extends beyond this specific disease.

### Challenges and Future Directions

The present work underscores the significance of associating feature space distributions with prediction scores for the purpose of developing an interpretable scoring system for the mortality prediction. One of the prevailing challenges within the medical domain pertains to the divergence between the development dataset and unseen cohorts, which poses a persistent issue for existing algorithms. In response to this challenge, we have introduced a normalization strategy tailored for out-of-distribution cohorts, which seeks to mitigate skewness, following the principles of the central limit theorem. Our proposed normalization technique necessitates the utilization of a representative cohort to ensure the reliability of outcomes. Furthermore, we have put forth a continuous normalization approach with instantaneous threshold adjustments; this, however, requires either a latency period or initial representative data for accurate normalization. Another challenge that need to be addressed is the application boundary of our approach. The application boundary is generally determined by the image quality as well as the cohort characterization of the development set. One of the foremost challenges lies in harmonizing and integrating multi-omics data, including transcriptomics, genomics, and epigenomics. Future research should focus on developing robust methodologies and computational tools to streamline such a process, including all available data types. Integrating multi-omics analysis into the clinical workflow is a significant challenge.

Future efforts will focus on validating our approach for clinical utility to optimize the treatment management for bladder cancer. Digital biomarkers, such as histomics, have the potential to serve as companion variables for disease staging and patient selection. Future research should also explore integration with Electronic Health Records (EHRs) and decision support systems, ensuring clinicians can access and utilize the integrated data efficiently. Integrating multi-omics data can further our understanding of disease mechanisms, potentially leading to breakthroughs in treatment and prevention. Yet, it is not clear whether omics strategies provide superior clinical benefits compared to a single data modality. Finally, possessing a scoring system that captures the omics features of the underlying disease from a single image modality (in our case, FFPE histology images) may help justify customizing the molecular profiling in the clinical setting.

## 5. Conclusions

Our scoring system has the potential to facilitate shared decision making by offering comprehensive and precise risk stratification, treatment planning, and cost-effective preselection for expensive molecular characterization.

## Figures and Tables

**Figure 1 cancers-15-04998-f001:**
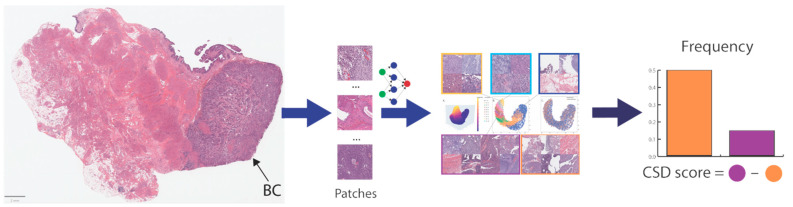
Illustrates the abstract AI framework for our approach. First, the tissue area is masked and tiled into small patches labeled with the cancer-specific survival status (weakly labeling). We trained the model to predict the cancer-specific survival status, and we then explored the distribution of the latent features and histology patterns stratified by the prediction deciles to develop the cancer-specific score system consisting of two prediction deciles (orange and lilac colors) reflecting distinguishable histology patterns.

**Figure 2 cancers-15-04998-f002:**
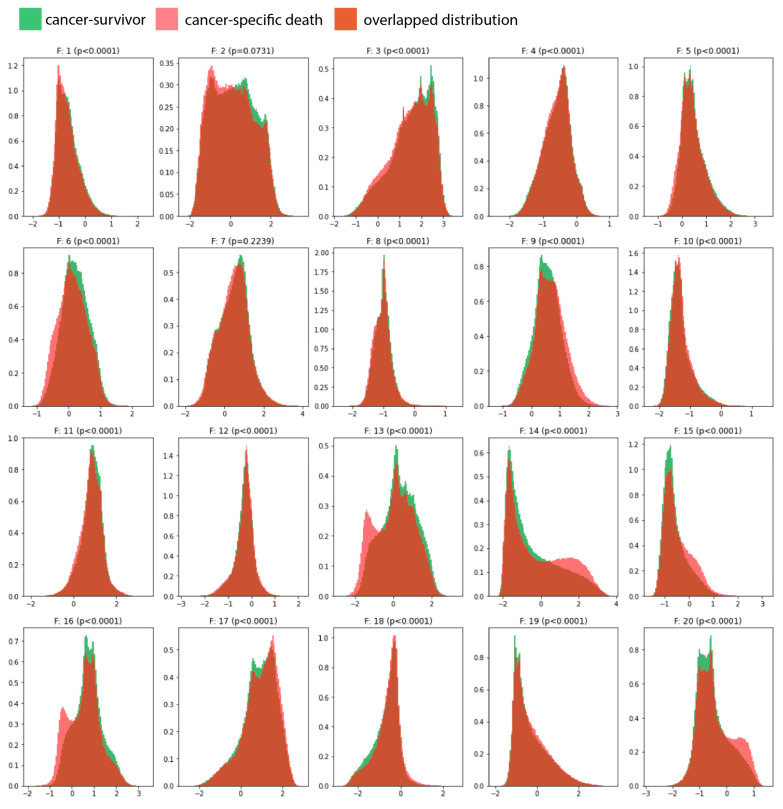
Lists the density histograms for the last 8 × 8 two-dimensional feature maps (pixels) according to the cancer-specific death status at the pixel level (i.e., pixel values) on 25,000 patches from the validation set. A size of 1 × 1 pixel on a feature map corresponds to an area with 64 × 64 pixels on the corresponding patch image (512 × 512 pixels). The Levene test was applied to assess the equality of variance between cancer survivors and cancer-specific death patches. We identified that some features (e.g., F13, F14, F15, F16, and F20) revealed histogram ranges for pixel values of specific feature maps more common in cancer-specific death patches (red areas).

**Figure 3 cancers-15-04998-f003:**
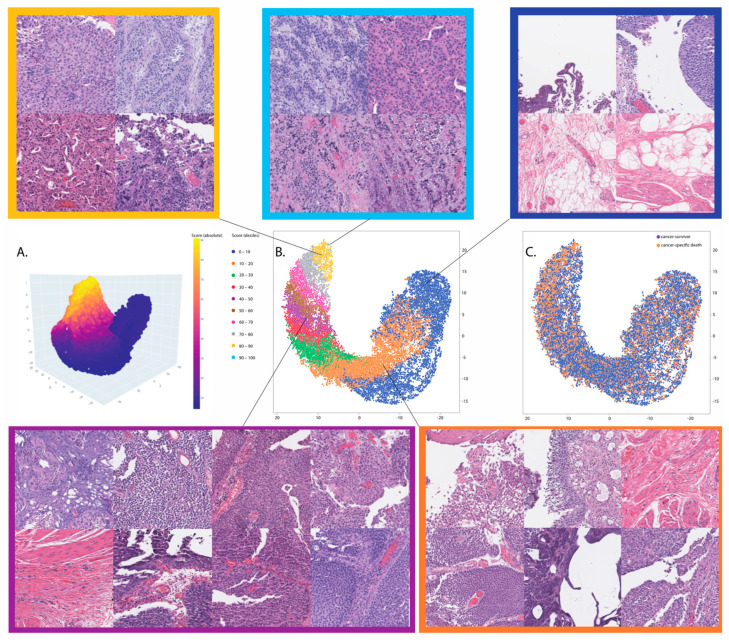
Summarizes the t-SNE visualization of penultimate features intuitively sorted by the deciles of the model inference scores (predications aka confidence) on representative 25,000 patch images randomly selected from the validation set. These patches represent the entire cases (n = 81) of the validation set. The corresponding patches were evaluated and identified to be altered by the prediction deciles. Based on the data evaluation and the domain knowledge, we selected the second decile and fifth decile; the second decile (orange color) was predominantly associated with negative patches (>50%), including bladder cancer, while the fifth decile (lilac color) was the center decile between the first and the ninth decile (the tenth decile was not considered due to its negligible sample size). (**A**) The 3D feature visualization; (**B**) the 2D visualization of features stratified by prediction deciles; (**C**) the 2D visualization of features stratified by the cancer-specific death status.

**Figure 4 cancers-15-04998-f004:**
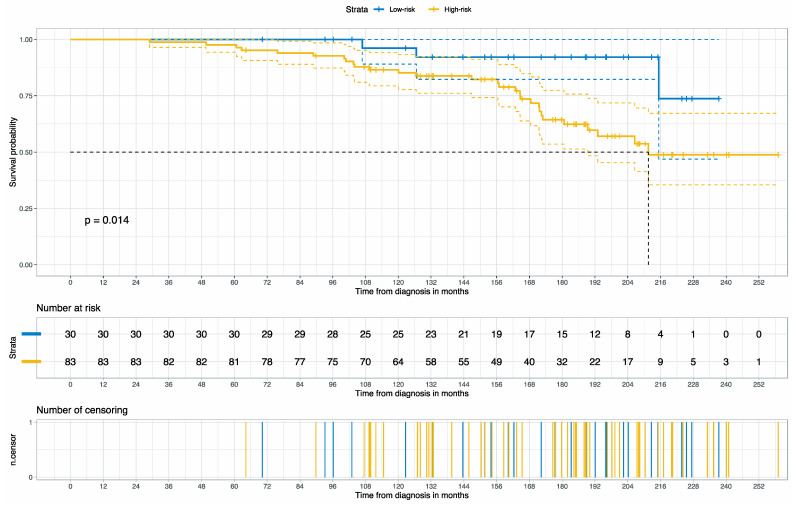
The Kaplan–Meier curve for cancer-specific survival stratified by the categorized risk scores (Low-risk vs. High-risk) on the PLCO validation cohort. The dot line reveals the median survival (the time it takes to reach 50% survival) between 210 and 216 months.

**Figure 5 cancers-15-04998-f005:**
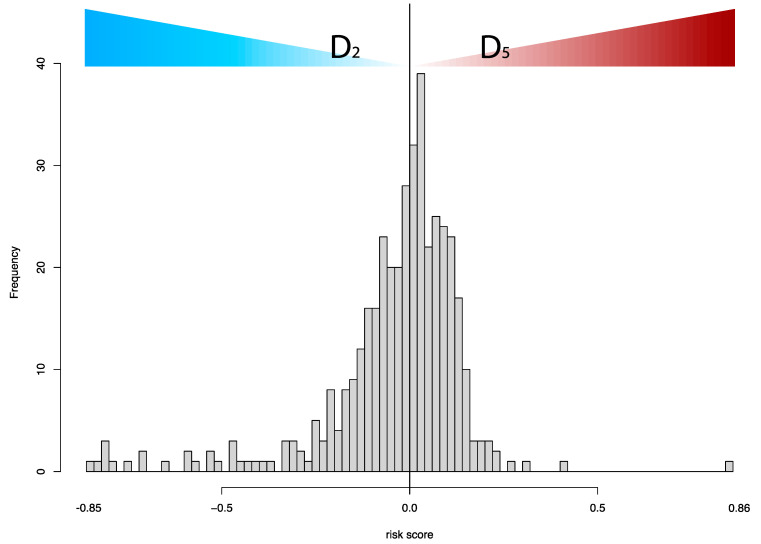
The distribution of the case risk scores is determined by D2 and D5, which are the relative frequencies of patches for the second and fifth deciles of the prediction for each case. The frequency corresponds to the case number. The cohort-specific threshold was estimated to be 0 for the TCGA dataset. Thresholding the risk scores results in two risk groups, where D2 and the high-risk group by D5 dominate the low-risk group. Figure 6 illustrates the histopathologic patterns associated with D2 and D5.

**Figure 6 cancers-15-04998-f006:**
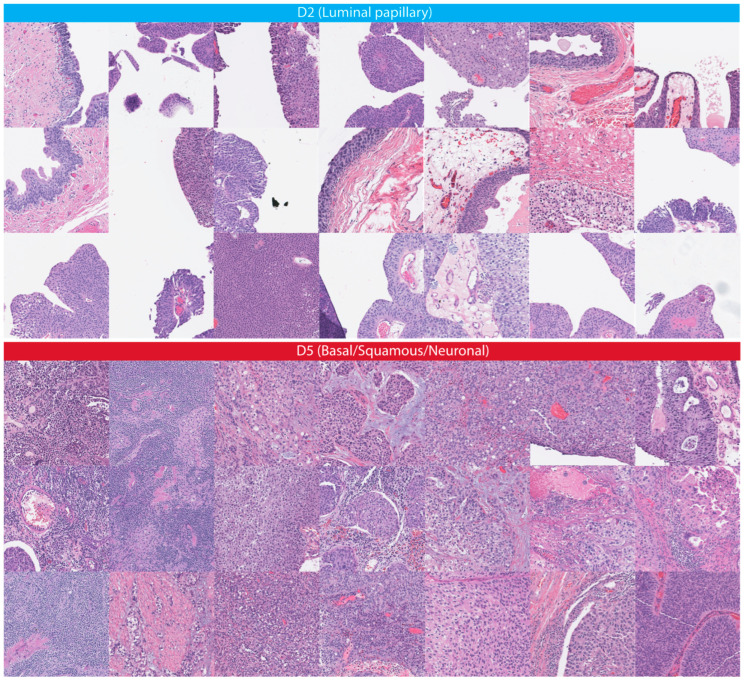
Exemplifies the distinct histopathologic patterns for D2 and D5 on the TCGA cohort. The absolute difference in the proportions between D2 and D5 in histology images determines whether the case is assigned to a low- or high-risk group (Figure 5). A negligible small fraction of patches in D2 solely included arteria vessels as luminal structures.

**Figure 7 cancers-15-04998-f007:**
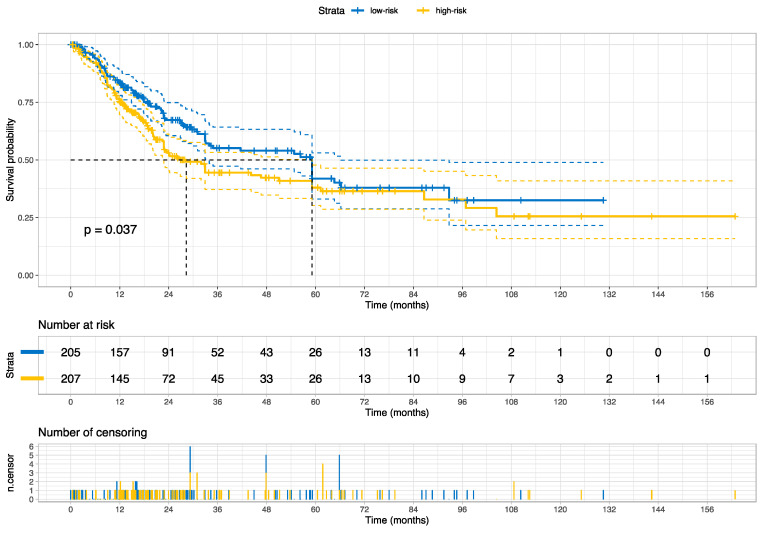
The Kaplan–Meier curve for overall survival stratified by the categorized risk scores (low-risk vs. high-risk) on the external validation set (TCGA cohort). *p* value was estimated using the log Rank test. The dot lines reveal the median survival for each risk group.

**Figure 8 cancers-15-04998-f008:**
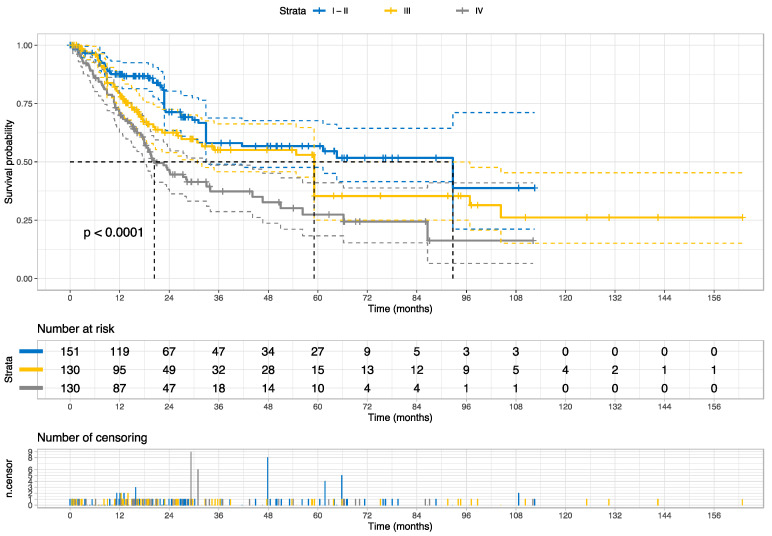
The Kaplan–Meier curve for overall survival stratified by the AJCC pathologic stages of bladder cancer on the external validation set (the TCGA Cohort). This staging system combines the subcategories of the TNM classification. *p* value was estimated using the log Rank test. The single case with unknown stage information was not visualized. The dot lines reveal the median survival.

**Figure 9 cancers-15-04998-f009:**
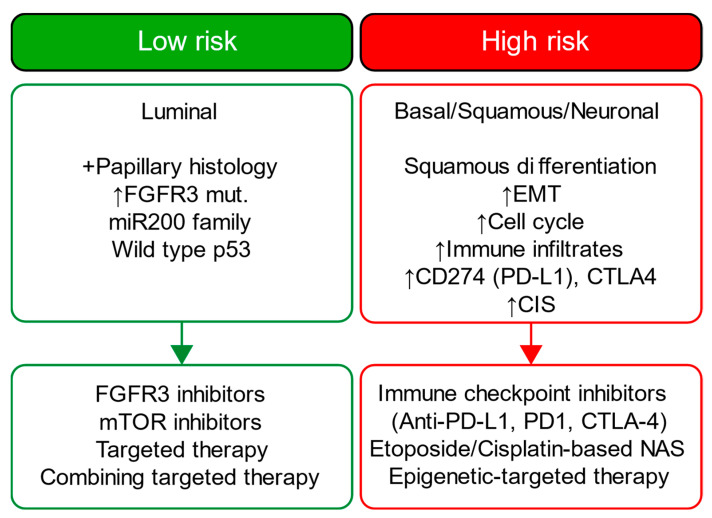
Overview of each risk group’s molecular characteristics and proposed treatment options. CIS: Carcinoma in situ; NAS: neoadjuvant chemotherapy; EMT: Epithelial–Mesenchymal Transition. The information is based on the TCGA-BLCA studies that investigated the treatment responses of main molecular subtypes (i.e., luminal, basal, squamous, and neural subtypes). We emphasize that this overview is abstract and not comprehensive and aims to generate hypotheses for potential treatment options for each risk group. The overview covers only the common main molecular subtypes (i.e., luminal and basal) for each risk group. The molecular features for these subtypes are already investigated by TCGA-BLCA studies.

**Table 1 cancers-15-04998-t001:** The search space for the neural architecture search.

Parameter	Options
Block architecture (microarchitecture design)	Inception block (Inception) Residual block (ResNet) Conventional block (VGG) Attention block (soft_att)
Width	2, 4, 6
Depth	3, 4, 5
Length (pathways)	2, 3, 4
Junctions (interconnection between pathways)	1, 2, 3
Global pooling	Average vs. Maximum
Addition of transformer	Yes vs. No

**Table 2 cancers-15-04998-t002:** The cohort description of the development set. ^+^Given the study’s history and design, the previous grade was available.

	Training Set	Optimization Set	Validation Set	
Characteristic	N = 26	N = 6	N = 81	*p*-Value ^1^
Age at diagnosis in year, median (IQR)	65.0 (62.2–68.8)	69.0 (65.8–70.0)	65.0 (61.0–68.0)	0.50
Sex, n (%)				0.13
Male	25 (96%)	5 (83%)	64 (79%)	
Female	1 (3.8%)	1 (17%)	17 (21%)	
WHO Grade 1973, n (%)^+^				0.26
G1	5 (19%)	1 (17%)	15 (19%)	
G2	10 (38%)	3 (50%)	15 (19%)	
G3	11 (42%)	2 (33%)	47 (58%)	
Unknown	0 (0%)	0 (0%)	4 (4.9%)	
AJCC tumor staging				
T stage, n (%)				0.35
Ta	4 (15%)	0 (0%)	1 (1.2%)	
Tis	11 (42%)	5 (83%)	31 (38%)	
T1	5 (19%)	1 (17%)	19 (23%)	
T2	3 (12%)	0 (0%)	22 (27%)	
T3	2 (7.7%)	0 (0%)	4 (4.9%)	
T4	0 (0%)	0 (0%)	1 (1.2%)	
Unknown	1 (3.8%)	0 (0%)	3 (3.7%)	
N stage, n (%)				0.46
Nx/N0	25 (96%)	6 (100%)	74 (91%)	
N1	0 (0%)	0 (0%)	4 (4.9%)	
Unknown	1 (3.8%)	0 (0%)	3 (3.7%)	
M stage, n (%)				0.47
Mx/M0	25 (96%)	6 (100%)	75 (93%)	
M1	0 (0%)	0 (0%)	3 (3.7%)	
Unknown	1 (3.8%)	0 (0%)	3 (3.7%)	
Follow-up duration in months, median (IQR)	172 (130–201)	151 (87–192)	168 (130–197)	0.80
Cancer-specific death, n (%)	6 (23%)	1 (17%)	25 (31%)	0.60
Whole-slide images, n (%)	46 (23.5%)	8 (4.1%)	142 (72.4%)	-
Patches, n (%)	26,949 (16.5%)	7574 (4.6%)	129,122 (78.9%)	-

^1^ Kruskal–Wallis rank sum test; Pearson’s Chi-squared test.

**Table 3 cancers-15-04998-t003:** The multivariate Cox regression analysis for cancer-specific mortality. HR: Hazard ratio; CI: Confidence Interval. Grading on the PLCO validation set. Due to the PLCO study design, only the WHO 1973 grading was available. Nonetheless, it is important to emphasize that WHO grading is a well-established prognostic parameter, lending significance to its inclusion in our analysis.

Variable	HR	95% CI	z	*p*
Age at diagnosis	1.03	(0.96–1.11)	0.87	0.39
Grading (WHO 1973)				
G1 (ref.)	–	–	–	–
G2	2.21	(0.20–24.48)	0.64	0.52
G3	11.99	(1.61–89.21)	2.43	0.02
Unknown	11.72	(1.03–133.02)	1.99	0.05
Risk score	8.39	(1.53–46.12)	2.45	0.01

**Table 4 cancers-15-04998-t004:** The cohort description of the external validation set.

Characteristic	N = 412
Age at diagnosis in years, median (IQR)	68 (60–76)
Sex, n (%)	
Female	107 (26%)
Male	305 (74%)
pM	
M0/x	398 (97%)
M1	11 (2.7%)
Unknown	3 (0.7%)
pN	
N0x	282 (68%)
M1	123 (30%)
Unknown	7 (1.7%)
pT	
T1	2 (0.5%)
T2	112 (27%)
T3	190 (46%)
T4	54 (13%)
Unknown	54 (13%)
Grade, n (%)	
Unknown	1 (0.2%)
High grade	390 (95%)
Low grade	21 (5.1%)
History of non-muscle invasive bladder cancer, n (%)	
Unknown	127 (31%)
NO	227 (55%)
YES	58 (14%)
Bladder cancer pathologic stage, n (%)	
I–II	151 (36.7%)
III	130 (31.6%)
IV	130 (31.6%)
Unknown	1 (0.2%)
Death, n (%)	185 (45%)
Follow-up duration in month, median (IQR)	19 (12–33)

**Table 5 cancers-15-04998-t005:** Multivariate Cox regression analysis for overall mortality. HR: Hazard ratio, CI: Confidence Interval. The AJCC pathologic tumor stage is a result of combining the subcategories of the TNM classification. We excluded the tumor grade as the muscle invasive bladder cancers are typically high-grade, and 95% of tumor grades in our cohort has high-grade BC.

Variable	HR	95% CI	z	*p*
High- vs. Low-risk group	1.35	(1.01–1.80)	1.99	0.0462
Age at diagnosis	1.02	(1.00–1.03)	2.32	0.0201
AJCC pathologic tumor stage				
I/II (ref.)	–	–	–	–
III	1.51	(1.03–2.21)	2.10	0.0357
IV	2.21	(1.54–3.18)	4.30	<0.0001

**Table 6 cancers-15-04998-t006:** Analysis summary of signatures and features associated with the risk group. *p*-values for a signature were estimated using Chi-Squared tests.

Signature	*p* Value	Features Associated with 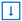 Low-Risk or 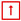 High-Risk Group
microRNA cluster	0.003998001	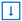 Cluster 3 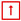 Cluster 1
mutation in TSC1	0.006496752	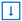 TSC1 mutation
mRNA cluster	0.009995002	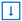 Luminal papillary 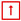 Basal/Squamous 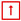 Neuronal
mutation in FGFR3	0.010994503	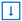 FGFR3 mutation
lncRNA cluster	0.012493753	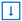 Cluster 3 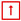 Cluster 4
mutation in ERBB3	0.016991504	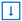 ERBB3 mutation
mutation in FAT1	0.023488256	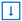 FAT1 mutation
mutation in PIK3CA	0.028485757	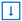 PIK3CA mutation
mutation in KANSL1	0.033983008	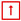 KANSL1 mutation
mutation in TMCO4	0.038480760	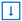 TMCO4 mutation
mutation in KDM6A	0.044977511	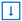 KDM6A mutation
mutation in METTL3	0.057971014	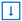 METL3 mutation
Squamous pathology	0.066466767	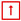 Squamous histopathology
mutation in PSIP1	0.075462269	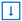 PSIP1 mutation
mutation in ZNF773	0.092453773	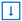 ZNF773 mutation
Hypomethylation cluster	0.092953523	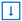 Cluster 4 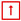 Cluster 2
mutation in GNA13	0.093953023	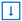 GNA13 mutation

**Table 7 cancers-15-04998-t007:** The distribution of molecular clusters for mRNA, lncRNA, miRNA, and DNA hypomethylation.

Risk Groups	Molecular Signatures	
mRNA		
	Luminal papillary	Basal/Squamous/Neuronal
Low-risk	85 (59%)	56 (36%)
High-risk	58 (41%)	101 (64%)
lncRNA		
	Cluster 3	Cluster 4
Low-risk	47 (64%)	61 (41%)
High-risk	26 (36%)	87 (59%)
miRNA		
	Cluster 3	Cluster 1
Low-risk	77 (62%)	30 (39%)
High-risk	47 (38%)	47 (61%)
DNA hypomethylation		
	Cluster 4	Cluster 2
Low-risk	23 (68%)	27 (39%)
High-risk	11 (32%)	42 (61%)

**Table 8 cancers-15-04998-t008:** The distribution of TSC1 mutation between the risk groups.

	TSC1 Gene	
Risk groups	wild-type	mutated
Low-risk	177 (47%)	28 (72%)
High-risk	196 (53%)	11 (28%)

**Table 9 cancers-15-04998-t009:** The distribution of ERBB3 mutation between the risk groups.

	ERBB3 Gene	
Risk groups	wild-type	mutated
Low-risk	175 (48%)	30 (67%)
High-risk	192 (52%)	15 (33%)

**Table 10 cancers-15-04998-t010:** The distribution of FGFR mutation between the risk groups.

	FGFR3 Gene	
Risk groups	wild-type	mutated
Low-risk	163 (47%)	42 (65%)
High-risk	184 (53%)	23 (35%)

## Data Availability

TCGA data and PCLO data are publicly available. The python jupyter notebook with python codes will be available at https://github.com/oeminaga/AI_BladderCancerMortality.git.

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
