# Peer review of "Artificial Intelligence Reveals Distinct Prognostic Subgroups of Muscle-Invasive Bladder Cancer on Histology Images"

_cancers, 2023, doi:10.3390/cancers15204998_

Round 1

Reviewer 1 Report

Dear authors,

Overall the manuscript is good. Please make the rechanges and resubmit it.

Abstract : Okay

Introduction: Mention research gaps properly. Also mention the novelty of your research. Literature review is weak . Deep learning has been used in various medical applications. Please add appropriate references. Following related articles could also be included.

1. Nayak T, Chadaga K, Sampathila N, Mayrose H, Muralidhar Bairy G, Prabhu S, Katta SS, Umakanth S. Detection of Monkeypox from skin lesion images using deep learning networks and explainable artificial intelligence. Applied Mathematics in Science and Engineering. 2023 Dec 31;31(1):2225698.

Methods:

Data: Add some images of the data

Images and some related tables must be added in the methods section

Results: One of the best sections of the paper.

Discussion: More comparison is required

Please add a seperate challlenges and future directions section.

 Please add a seperate conclusion section.

Author Response

Overall, the manuscript is good. Please make the rechanges and resubmit it.

Abstract : O.K.

Introduction: Mention research gaps properly. Also mention the novelty of your research. Literature review is weak . Deep learning has been used in various medical applications. Please add appropriate references. Following related articles could also be included.

  1. Nayak T, Chadaga K, Sampathila N, Mayrose H, Muralidhar Bairy G, Prabhu S, Katta SS, Umakanth S. Detection of Monkeypox from skin lesion images using deep learning networks and explainable artificial intelligence. Applied Mathematics in Science and Engineering. 2023 Dec 31;31(1):2225698.

Authors’ reply:

We thank the reviewer for the feedback. Accordingly, we added the following references and the paragraph in the introduction section to improve the literature overview in the introduction.

The paragraph, page 2, lines 64 – 68:

“For instance, Woerl et al. reported the potential of DL to forecast the molecular subtypes of MIBC by analyzing hematoxylin and eosin (H&E) slides [7]. As a proof-of-concept, Mundhada et al. have shown the DL capability to distinguish low-grade from high-grade histology [8]. Zheng et al. purposed a DL  framework to predict survival from histology images with BC [9].”

Methods:

Data: Add some images of the data

Images and some related tables must be added in the methods section.

Authors’ reply:

Accordingly, we added Fig 1. as merit illustration of a WSI image of our dataset as well as we provided an abstract illustration of the AI algorithm. In addition, we added Table 1 that include the search space for neural architecture search.

Results: One of the best sections of the paper.

Authors’ reply:

We appreciate the positive reviewer’s comment.

Discussion: More comparison is required

Authors’ reply:

We already provided the following paragraph in the discussion section, page 18, lines 452 - 472:

“Earlier studies in bladder cancer applied deep learning to infer staging [31], grade [32, 33], recurrence risk [34], FGFR3 mutation [35], and specific molecular subtypes [7] from histology images. Although some previous studies examined the prediction of molecular targets, the current study found that prognostic histopathological patterns for bladder cancer are rather associated with multi-omics profiles (i.e., transcriptomic, genomics, and epigenomics); these multi-omics profiles are already covering the specific molecular subtypes and the FGFR3 mutations investigated earlier, and we have shown that the accuracy of our risk groups for FGFR3 mutation is similar to the previous report, signifying the impact of multi-omics profiles as confounding factors on the results of earlier studies. In support of our findings, the BLCA-TCGA study (molecular characterization of bladder cancer) revealed that the molecular subtypes and signatures are linked with each other and distinct histopathologic patterns (e.g., papillary, basal/squamous) were connected with omics profiles that are prognostic and have different therapeutic targets [3, 17]. A comparable study in Lung cancer reported that omics features are predictive of histology patterns as well [36]

Although multiple studies identified the detection potential of single mutations or specific molecular subtypes from histology images [37-41], the histopathological appearance is mainly driven by a collection of multifaceted molecular modulations and reflects the cancer malignancy and survival. Subsequently, establishing a direct association between a single molecular signature and histology images must be inadequate, given other confounders for bladder cancers.”

We would like to emphasize that discussing c-index from different literature as comparison metric for survival predictions is not appropriate due to the metric limitation and the differences in the test datasets.

Please add a separate challenges and future directions section.

Authors’ reply:

Accordingly, we added the following paragraph in the discussion section, page 21, line 584 – 613:

“Challenges and Future Directions

The present work underscores the significance of associating feature space distributions with prediction scores for the purpose of developing an interpretable scoring system for the mortality prediction. One of the prevailing challenges within the medical domain pertains to the divergence between the development dataset and unseen cohorts, which poses a persistent issue for existing algorithms. In response to this challenge, we have introduced a normalization strategy tailored for out-of-distribution cohorts, which seeks to mitigate skewness, following the principles of the central limit theorem. Our proposed normalization technique necessitates the utilization of a representative cohort to ensure the reliability of outcomes. Furthermore, we have put forth a continuous normalization approach with instantaneous threshold adjustments;  this, however, requires either a latency period or initial representative data for accurate normalization. Another challenges that need to be addressed is the application boundary of our approach. The application boundary is generally determined by the image quality as well as the cohort characterization of the development set. One of the foremost challenges lies in harmonizing and integrating multi-omics data, including transcriptomics, genomics, and epigenomics. Future research should focus on developing robust methodologies and computational tools to streamline such process including all available data types. Integrating multi-omics analysis into the clinical workflow is a significant challenge.

Future efforts will focus on validating our approach for the clinical utility to optimize the treatment management for bladder cancer. Digital biomarkers, such as histomics, have the potential to serve as companion variables for disease staging and patient se-lection. Future research should also explore integration with Electronic Health Records (EHRs) and decision support systems, ensuring clinicians can access and utilize the integrated data efficiently. Integrating multi-omics data can further our understanding of disease mechanisms, potentially leading to breakthroughs in treatment and prevention. It is not clear whether omics strategies provide superior clinical benefits compared to a single data modality. Finally, possessing a scoring system that captures the omics features of the underlying disease from a single image modality (in our case, FFPE histology images) may help justify customizing the molecular profiling in the clinical setting”

Please add a separate conclusion section.

Authors’ reply:

“Our scoring system has the potential to facilitate shared decision-making by offering comprehensive and precise risk stratification, treatment planning, and cost-effective preselection for expensive molecular characterization.“

Reviewer 2 Report

In this study, the authors developed an AI-based data scoring system to better investigate bladder cancer histology images. They used a sufficient amount of sample data to validate their system. Overall this manuscript is good in written and well demonstrating the points.

Some improvement could be done, such as in lines 38 and 41, two "bladder cancer (BC)" appeared; in line 44, no explanation for BCA.

Author Response

In this study, the authors developed an AI-based data scoring system to better investigate bladder cancer histology images. They used a sufficient amount of sample data to validate their system. Overall, this manuscript is good in written and well demonstrating the points.

Some improvement could be done, such as in lines 38 and 41, two "bladder cancer (BC)" appeared; in line 44, no explanation for BCA.

Authors’ reply:

We would like to thank the reviewer for the positive feedback. Accordingly, we removed the duplicate "bladder cancer (BC)" and corrected BCA to BC.

Reviewer 3 Report

In general the manuscript was well performed. The ML application in this field is with high level of interest. The explainability of methods are also writed and conducted in a good way. Muscle invasive bladder cancer (MIBC) is a highly heterogeneous and costly disease with 15 significant morbidity and mortality. Understanding tumor histopathology leads to tailored therapies and improved outcomes. In this study, the authors employed weakly supervised learning and neural 17 architecture search to develop a data-driven scoring system. About the introduction, i suggest to improve tne number of works with similar outcomes. If there are low number of paper on this pathology, the authors can manage other similar pathologies in order to improve the intoduction for the readers. In the same time references should be improved with focus on similar models and applicability in real context including research and clinics evidences. Methods are well conducted and exposed in a good way. I suggest to stress the limitations related to the low number of occurrences in the model test and validation. Another point is related to the multifactorial analyses needed to extract relationship between studied covariates in the external cohort validation. What is the impact of any one of these? what is the relationship between the included covariates (to control for autocorrelation). Step-wise or lasso models can support this evaluation to select a cut-off of covariates needed for the model AUC evaluation (BIC and AIC should be reported). Moving to the discussion the general suggestion is to stress the potential clinical applications, the next steps in research in the same or application in other disease. What is the value of the work, what is the impact? Editing is needed to simplify some parts and facilitate the reading. Please stress the multi omics approaches overview as a next step, considering potential issues in the not simple clinical applications. 

thx

Editing is needed in all parts to boost the simplicity of the work including grammar and form.

Author Response

Reviewer #3

In general, the manuscript is well written. The ML application in this field is with a high level of interest. The explainability of methods are also written and conducted in a good way. Muscle invasive bladder cancer (MIBC) is a highly heterogeneous and costly disease with 15 significant morbidity and mortality. Understanding tumor histopathology leads to tailored therapies and improved outcomes. In this study, the authors employed weakly supervised learning and neural 17 architecture search to develop a data-driven scoring system.

About the introduction, I suggest improving the number of works with similar outcomes. If there are low number of papers on this pathology, the authors can manage other similar pathologies in order to improve the introduction for the readers. At the same time, references should be improved with focus on similar models and applicability in real context including research and clinics evidence.

Authors’ reply:

We thank the reviewer for the feedback. Accordingly, we added the following references and the paragraph in the introduction section to improve the introduction.

The paragraph, page 2, lines 64 – 68:

“For instance, Woerl et al. reported the potential of DL to forecast the molecular subtypes of MIBC by analyzing hematoxylin and eosin (H&E) slides [7]. As a proof-of-concept, Mundhada et al. have shown the DL capability to distinguish low-grade from high-grade histology [8]. Zheng et al. purposed a DL  framework to predict survival from histology images with BC [9].”

Methods are well conducted and exposed in a good way.

1) I suggest stressing the limitations related to the low number of occurrences in the model test and validation.

Authors’ reply:

We thank the reviewer for the feedback. The cancer-specific death in the internal validation set was 31% and the overall death rate for the external validation set was 45%, which are fairly enough number to derive reasonable outcome. The goal of the optimization set is to select the model with the best performance at the patch level and we believe it is succinct to select the appropriate model given the large sample size at patch level. We emphasize that the training set, the optimization, and validations sets are originated from non-overlapping institutions. Our event per variable (EPV) in the external validation phase meets the criteria set by Ogundimu et al.; our study does not incorporate low-prevalence predictors in our model, and we have approximately 10 events per variable (EPV), which aligns with the widely accepted rule of thumb (PMID: 26964707).

2) Another point is related to the multifactorial analyses needed to extract relationship between studied covariates in the external cohort validation. What is the impact of any one of these? what is the relationship between the included covariates (to control for autocorrelation). Stepwise or lasso models can support this evaluation to select a cut-off of covariates needed for the model AUC evaluation (BIC and AIC should be reported).

Authors’ reply:

We thank the reviewer for suggesting the multifactorial analyses on the external cohort validation to extract relationship between covariates (age at diagnosis, scores, and pathologic tumor stages). We replicated the scheme analyses conducted by the original TCGA study for bladder cancer. The multivariate Cox regression analyses on the external validation set show the independency of each variable for overall mortality which is the goal of the analyses. We applied VIF (variance inflation factor) to assess the Multicollinearity in the COX regression model. Here, we found that VIFs of all variables were below <2, indicating that the multicollinearity is negligible small. Since our goal is not to select features in the Cox regression model, BIC and AIC are not informative. Accordingly, we added the following sentences in method and result sections:

Method section, page 6, lines 263 and 264:

“VIF (variance inflation factor) was used to assess the multicollinearity in the COX regression model [28].”

Result section, page 12, line 363-364:

“The multicollinearity for these covariates was negligible small (VIFs < 2).”

Moving to the discussion:

the general suggestion is to stress the potential clinical applications, the next steps in research in the same or application in other disease. What is the value of the work, what is the impact? Editing is needed to simplify some parts and facilitate the reading.

Please stress the multi omics approaches as a next step, considering potential issues in the not simple clinical applications. 

Authors’ reply:

Accordingly, we added the following paragraph in the discussion section, page 21, lines 584 - 613:

“Challenges and Future Directions

The present work underscores the significance of associating feature space distributions with prediction scores for the purpose of developing an interpretable scoring system for the mortality prediction. One of the prevailing challenges within the medical domain pertains to the divergence between the development dataset and unseen cohorts, which poses a persistent issue for existing algorithms. In response to this challenge, we have introduced a normalization strategy tailored for out-of-distribution cohorts, which seeks to mitigate skewness, following the principles of the central limit theorem. Our proposed normalization technique necessitates the utilization of a representative cohort to ensure the reliability of outcomes. Furthermore, we have put forth a continuous normalization approach with instantaneous threshold adjustments;  this, however, requires either a latency period or initial representative data for accurate normalization. Another challenges that need to be addressed is the application boundary of our approach. The application boundary is generally determined by the image quality as well as the cohort characterization of the development set. One of the foremost challenges lies in harmonizing and integrating multi-omics data, including transcriptomics, genomics, and epigenomics. Future research should focus on developing robust methodologies and computational tools to streamline such process including all available data types. Integrating multi-omics analysis into the clinical workflow is a significant challenge.

Future efforts will focus on validating our approach for the clinical utility to optimize the treatment management for bladder cancer. Digital biomarkers, such as histomics, have the potential to serve as companion variables for disease staging and patient se-lection. Future research should also explore integration with Electronic Health Records (EHRs) and decision support systems, ensuring clinicians can access and utilize the integrated data efficiently. Integrating multi-omics data can further our understanding of disease mechanisms, potentially leading to breakthroughs in treatment and prevention. It is not clear whether omics strategies provide superior clinical benefits compared to a single data modality. Finally, possessing a scoring system that captures the omics features of the underlying disease from a single image modality (in our case, FFPE histology images) may help justify customizing the molecular profiling in the clinical setting.”